# Application of Fiber Bragg Grating Acoustic Emission Sensors in Thin Polymer-Bonded Explosives

**DOI:** 10.3390/s18113778

**Published:** 2018-11-05

**Authors:** Tao Fu, Peng Wei, Xiaole Han, Qingbo Liu

**Affiliations:** 1Institute of Chemical Materials, China Academy of Engineering Physics, Mianyang 621900, China; futao@caep.cn; 2School of Instrumentation and Optoelectronic Engineering, Beihang University, Beijing 100191, China; hanxiaole@buaa.edu.cn (X.H.); liuqingbo@buaa.edu.cn (Q.L.)

**Keywords:** fiber Bragg grating acoustic emission sensor, directionality, time coefficient location method, polymer-bonded explosives

## Abstract

Fiber Bragg grating (FBG) acoustic emission (AE) sensors have been used in many applications. In this paper, based on an FBG AE sensor, the sensing principle of the interaction between the AE wave and the sensor is introduced. Then, the directionality of the FBG AE sensor on the surface of a thin polymer-bonded explosive (PBX) material is studied. Finally, the time coefficient location method is proposed to correct the AE time detected by the FBG AE sensor, thereby improving the accuracy of location experiments.

## 1. Introduction

Polymer-bonded explosive (PBX) materials are popular materials for use in weapons because of their advanced high-energy performance. To safely store a PBX in a narrow container, a thin PBX is produced; however, when the environmental temperature changes, cracks can form on this critical material because of its low thermal conductivity. As a result, the risk of explosion of the thin PBX material will increase. Therefore, a stable and reliable nondestructive testing method is urgently needed to monitor the health of thin PBX materials.

Acoustic emission (AE) detection is a nondestructive testing method. Since the initial development of AE theory in the 1950s, AE detection has been successfully used in the aerospace, machinery manufacture, petrochemical industries and other fields [1]. Currently, a piezoelectric (PZT) sensor, which is driven by electricity, is the only commercially available AE sensor [2]. However, a PZT sensor is not appropriate for use with a thin PBX material, as shown in Figure 1. In Figure 1, for the thin PBX material, the diameter is 220 mm, and the height is 20 mm; thus, the gap between the container and the thin PBX is 1 mm, which is too narrow to allow a PZT AE sensor and its shielded cable to be affixed. Moreover, the PZT sensor and its cable can generate heat or sparks or short-circuit during operation, all of which pose danger to the PBX material. In Figure 1, the diameter is much larger than the height, so in this paper, the thin PBX can be considered as a 2-dimensional model.

Since the 1970s, various optical fiber sensors and signal demodulation methods have been introduced in many fields [3]. In the civil infrastructure monitoring field, optical fiber sensors can be used to detect temperature, strain, pressure, vibration, current, voltage, and gas [4]. In the biomedical field, optical fiber sensors are used for ultrasound ablation, distributed catheter shape sensing, wearable sensors, and smart textiles [5]. In materials science, optical fiber sensors are used to detect vibration, bending, and torsion [6].

The optic fiber AE sensor is a new kind of AE sensor that has a small size, light weight, high sensitivity, anti-electromagnetic interference capability, and corrosion resistance. The fiber-optic AE sensor and its fiber pigtail do not introduce any heat or sparks during operation. These features would be large advantages in thin PBX testing.

At present, there are four kinds of fiber-optic AE sensor technologies. One type is fiber-optic F-P cavity AE technology, which has a wide measurement range and a stable system but is also difficult to produce and expensive [7]. Another type is fiber-optic ring AE technology, which has higher sensitivity than that of the other types [8,9] but also has a large fiber-optic ring [10,11]. The third type is fiber-optic coupler AE technology, but the sensor itself is too frail to use in practice [12,13]. The final type is the FBG AE sensor, which is a normal FBG sensor used for AE wave detection. The FBG AE sensor is the smallest among all types of fiber-optic AE sensors and is suitable for use in the narrow gap shown in Figure 1 [14,15]. This type of sensor can be easily produced and is low cost. Currently, research on FBG AE sensor technology focuses on detection sensitivity [16,17], frequency response range [18,19], and crack location [20,21].

Figure 2 shows the difference between the FBG AE sensor and the PZT AE sensor. Figure 2a shows the smallest commercial PZT AE sensor; its height is 2.4 mm, and its diameter is 3.6 mm. The AE wave, which is transient elastic wave, arrives at the bottom of the PZT AE sensor and oscillates inside the PZT AE sensor. During the oscillation, the AE wave will cause the PZT effect. Figure 2b shows an FBG AE sensor; its length is 10 mm, and its diameter is 0.25 mm. The AE wave passes through the FBG grid and generates strain in the FBG AE sensor. Based on Figure 1 and Figure 2, the FBG AE sensor is clearly smaller and more suitable for the gap with a thin PBX than the PZT AE sensor.

Thus, it can be concluded that the FBG AE sensor is more suitable for thin PBX materials than the other types of AE sensors.

## 2. Directionality of the FBG Sensor

The central wavelength of the FBG sensor is described in (1):(1)$$\lambda_{B}=2n\Lambda \;$$ where *λ**_B_* is the FBG sensor central wavelength, *n* is the effective refractive index, and Λ is the grating period.

When an AE wave arrives at the FBG sensors, it will change *n*, Λ and strain *ε*, as shown in Figure 3. In Figure 3, the coordinate system is established. The coordinate origin O is the center of the FBG sensor. The *Y* axis is along the axial direction of the FBG sensor. The *X* axis is perpendicular to the *Y* axis. *ε* is decomposed into *ε_x_* and *ε_y_*. *θ* is the angle between the AE wave propagation direction and the FBG sensor axial direction. *θ* is from 0° to 90° [22].

The quantities *n* and Λ are described in (2) and (3), respectively:(2)$$n=n_{0}-\frac{n_{0}^{3}}{2}[P_{12}\varepsilon_{y}+(P_{11}+P_{12})\frac{\varepsilon_{x}}{2}]\;$$
(3)$$\Lambda =\left(1+\varepsilon_{y}\right)\Lambda_{0}\;$$
where *n*_0_ is the initial effective refractive index, Λ_0_ is the initial grating period, and *P*_11_ and *P*_12_ are Pockel’s strain-optic coefficients [23].

Based on (1), Δ*λ_B_* can be given as:(4)$$\Delta \lambda_{B}=\left(\frac{\Delta n}{n_{0}}+\frac{\Delta \Lambda }{\Lambda_{0}}\right)\cdot \lambda_{B}\;$$ where Δλ_B_ is the shift in the FBG sensor central wavelength, ∆*n* is the shift in the effective refractive index, and ΔΛ is the shift in the grating period.

When the AE wave is propagating along the *Y* axis, *ε_x_* and *ε_y_* can be given as:(5)$$\varepsilon_{x}=-\mu \varepsilon_{y}\;$$ where *μ* is the Poisson ratio of the fiber.

Based on (1)–(5), Δ*λ_B_*_1_ is obtained by:(6)$$\begin{array}{ll} \Delta \lambda_{B}{}_{1} & =\left(\frac{\Delta n}{n_{0}}+\frac{\Delta \Lambda }{\Lambda_{0}}\right)\lambda_{B}\\ & =\left(\frac{n-n_{0}}{n_{0}}+\frac{\Lambda -\Lambda_{0}}{\Lambda_{0}}\right)\lambda_{B}\\ & =\left\{1-\frac{n_{0}^{2}}{2}\left[P_{12}-\frac{\mu }{2}\left(P_{11}+P_{12}\right)\right]\right\}\lambda_{B}\varepsilon_{y}\\ & =C_{y}\lambda_{B}\varepsilon_{y}\\ & =C_{y}\lambda_{B}\varepsilon \cdot \cos\theta \end{array}$$ where Δ*λ_B_*_1_ is the shift in the FBG sensor central wavelength when the AE wave is propagating along the *Y* axis and $C_{y}=1-\frac{n_{0}^{2}}{2}\left[P_{12}-\frac{\mu }{2}\left(P_{11}+P_{12}\right)\right]$. The shift in the FBG sensor central wavelength is caused by strain generated by the AE wave.

When the AE wave is propagating along the *X* axis, *ε_x_* and *ε_y_* can be given as:(7)$$-\mu \varepsilon_{x}=\varepsilon_{y}\;$$

Based on (1)–(4) and (7), Δ*λ_B_*_2_ is obtained by
(8)$$\begin{array}{ll} \Delta \lambda_{B2} & =\left\{-\mu -\frac{n_{0}^{2}}{2}\left[-\mu P_{12}+\frac{1}{2}\left(P_{11}+P_{12}\right)\right]\right\}\lambda_{B}\varepsilon_{x}\\ & =C_{x}\lambda_{B}\varepsilon_{x}\\ & =C_{x}\lambda_{B}\varepsilon \cdot \sin\theta \end{array}$$
where Δ*λ_B_*_2_ is the shift in the FBG sensor central wavelength when the AE wave is propagating along the *X* axis and $C_{x}=-\mu -\frac{n_{0}^{2}}{2}\left[-\mu P_{12}+\frac{1}{2}\left(P_{11}+P_{12}\right)\right]$.

When the AE wave is propagating in an arbitrary direction, Δ*λ_B_* can be decomposed into Δ*λ_B_*_1_ and Δ*λ_B_*_2_. Based on Figure 3 and (6) & (8), Δ*λ_B_* is obtained by:(9)$$\begin{array}{ll} \Delta \lambda_{B} & =\Delta \lambda_{B1}+\Delta \lambda_{B2}\\ & =(C_{y}\cos\theta +C_{x}\sin\theta )\lambda_{B}\varepsilon \end{array}$$

Based on (9), Δ*λ_B_* is clearly related to *θ*.

For an FBG sensor fabricated from standard silica fiber, *n*_0_ = 1.45, *μ* = 0.17, *P*_11_ = 0.12, and *P*_12_ = 0.275. Thus, *C_x_ = −0.328*, *C_y_ = 0.746*, *λ_B_ > 0*, and *ε*
*> 0*. Therefore, based on (9), the maximum value of Δ*λ_B_* is reached when *θ =* 0*°*, and the minimum value of Δ*λ_B_* is reached when *θ =* 90*°.*

The tunable narrowband laser detection method is used to demodulate the AE signal [24]. Figure 4 shows the tunable narrowband laser detection system and its illustrative diagram.

In Figure 4a, the tunable narrowband light penetrates the optical circulator and enters the FBG sensor. The reflected light of the FBG sensor containing the AE wave signal enters the photodetector and is then amplified by the preamplifier before finally reaching the acquisition system.

In Figure 4b, the bandwidth of the tunable narrowband laser is 0.0016 p.m. The bandwidth of the FBG sensor is 200 p.m., which is far greater than the tunable narrowband laser bandwidth. Thus, the spectrum of the tunable narrowband laser can be drawn as a line, as shown in Figure 4b. In the beginning, the output of the tunable narrowband laser is adjusted to the location of the steepest slope near the 3 dB point of the FBG sensor, for example, the C1 point in Figure 4b, and the reflected light power of the FBG sensor is denoted C1C2. When the AE wave arrives at the FBG sensor, the reflected light power of the FBG sensor will change to A1A2 or B1B2. After the photodetector and the preamplifier, the output voltage of the acquisition system is proportional to A1A2 or B1B2 and is related to *θ*, i.e., the directionality of the FBG sensor. In Figure 4, the tunable narrowband laser is the key device because it would be adjusted to fit the different FBG sensors.

The measurement results of the tunable narrowband laser detection method are dependent on spectral distortions of the attached gratings. In this paper, the spectra of the gratings show that the side-mode suppression ratio of the FBG sensor is 20 dB, the reflectivity of the FBG sensor is 96%, and the bandwidth of the FBG sensor is 200 p.m. These three parameters include an assessment of the quality of the spectra of the gratings.

## 3. Experiments and Results

### 3.1. Experimental Device

The four-channel FBG AE sensor system is shown in Figure 5, in which the tunable narrowband laser light is divided into L1, L2, L3 and L4 by coupler 1, coupler 2 and coupler 3. The power ratio of coupler 1, coupler 2 and coupler 3 is 50:50. L1 penetrates circulator 1 and enters FBG AE sensor 1. The reflected light with the AE wave signal from FBG AE sensor 1 is detected by photodetector 1, and then the signal is amplified by preamplifier 1 and reaches the acquisition system. L2, L3, and L4 are identical to L1. In Figure 5, four FBG AE sensors are chosen to have the same 3 dB point and temperature characteristics; i.e., the four sensors exhibit the same central wavelength shift when the temperature changes. The four FBG AE sensors and the FBG temperature sensor are placed in the same temperature region. The FBG temperature sensor is used to measure the temperature changes. When the temperature changes, the signal from the FBG temperature sensor is sent to the temperature demodulator system and the computer. Thus, when the temperature changes, the computer can adjust the output of the tunable narrowband laser to the location of the steepest slope near the 3 dB point of the four FBG AE sensors, as shown in Figure 4b.

Based on Figure 5, the four-channel FBG AE sensor system is built, as shown in Figure 6. The tunable narrowband laser is model TSL-510 from Santec Corporation (Aichi, Japan); the output wavelength is 1550 nm, and the power is 8 mW. The photodetector is model 2117 from New Focus Company (Irvine, CA, USA). The preamplifier is model 2/4/6 from Physical Acoustics Corporation (Princeton Junction, NJ, USA). The acquisition device is model Express 8 from Physical Acoustics Corporation. The computer is model IPC-610L from ADVANTECH Company (Taiwan, China). The temperature demodulator system includes model FBGA-S-1525-1605-FA from BaySpec Corporation (San Jose, CA, USA) and model ASE-CL-M2 from Top Photonics Corporation (Beijing, China).

### 3.2. FBG AE Sensor Directionality Experiment

The FBG AE sensor is glued onto the surface of a thin PBX with cyanoacrylate, as shown in Figure 7.

Figure 7a shows a photograph of the actual FBG AE sensor setup. In Figure 7a, the FBG AE sensor whose length is 10 mm is pasted along the horizontal direction. Along the horizontal directions, the angles are 0° and 180°. Along the perpendicular directions, the angles are 90° and 270°.

The Hsu-Nielsen broken lead method, is used in the experiments, where the pencil lead diameter is 0.5 mm, the hardness is 2H and the length is 3.0 mm. When this kind of pencil lead is broken on the surface of PBX materials, it will produce a standard AE signal. That signal can be used to simulate the AE signal at the breaking lead position [25,26]. For example, when the broken lead is at position A in Figure 7b, it will simulate an AE signal conducted at position A.

Figure 7b shows the location of broken lead points. In Figure 7b, the broken lead points are located at 36 different directions: 0°, 10°, 20°, 30°, 40°, …, 330°, 340°, and 350°. In each direction, the distance between the center of the FBG AE sensor and the broken lead points is 30 mm. The pencil lead is broken three times at each broken lead point. Using one channel of the FBG AE sensor system, as shown in Figure 5, the threshold voltage of the acquisition system is set to 100 µV (40 dB), and then the signal amplitude and arrival time *T* are obtained. The average signal amplitude of three broken leads at each point is shown in Figure 8.

In Figure 8, the dot-dashed line shows the angle from 0° to 330°, and the dashed line shows the amplitude from 40 dB to 80 dB. The red round dot is the signal amplitude along the directions of 0°, 10°, 20°, 30°, 40°, …, 330°, 340°, and 350°.

In Figure 8, by connecting the red round dots, the blue curve is obtained. Clearly, the blue curve exhibits longitudinal and horizontal symmetry. Thus, the curve from 0° to 90° can be studied as an example. Moreover, other areas are similar to it.

In Figure 8, the signal amplitude is clearly dependent on angle *θ*. The highest signal amplitude is observed along the 0° direction, and the lowest amplitude is observed along the 90° direction. Therefore, it verified (9) and the directionality of the FBG AE sensor.

The average arrival time *T* of three broken leads in the range from 0° to 90° is shown in Table 1.

Based on Table 1 and the curve fitting tool in MATLAB software, the relationship between angle *θ* and time *T* is given as:(10)$$T(\theta )=-2.623\times 10^{-8}\cdot \theta^{4}+1.429\times 10^{-5}\cdot \theta^{3}-4.542\times 10^{-4}\cdot \theta^{2}+1.314\times 10^{-2}\cdot \theta +2.3\;$$ where *T*(*θ*) is a 4th order polynomial function. Based on (10) and Table 1, Figure 9 can be obtained.

In Figure 9, the black points denote the real arrival time based on Table 1, and the blue curve is *T*(*θ*) based on (10); the R-square between them is 0.9936. Thus, the blue curve is found to match the black points very well. The 4th-order polynomial function *T*(*θ*) may not be the best one, but in this paper, it works well.

In Figure 9, the signal arrival time increases with the angle, thus verifying the directionality of the FBG AE sensor. Based on traditional AE location theory, the signal arrival time determines the location [27]. In other words, the directionality of the FBG AE sensor will influence the location.

### 3.3. AE Source Location Experiment on a Thin PBX Sample

Figure 10a shows a photograph of the sensors on the setup and the angles for the source location experiment. In Figure 10a, FBG AE sensor 1 and sensor 3 are pasted along the horizontal direction, and sensor 2 and sensor 4 are pasted along the perpendicular direction. Along the horizontal directions, the angles are 0° and 180°. The 45° direction is between the 0° and 90° directions. Moreover, along the perpendicular directions, the angles are 90° and 270°. The 225° direction is between the 180° and 270° directions.

Figure 10b shows the FBG AE source location experiment on a thin PBX sample and the corresponding source location experiment. In Figure 10b, there are four FBG AE sensors: sensor 1, sensor 2, sensor 3 and sensor 4. The offset between the center of sensor 1 and the *X* axis is 50 mm, and the offset between the center of sensor 2 and the *Y* axis is 50 mm; the same is true for sensor 3 and sensor 4. *θ*_1_, *θ*_2_, *θ*_3_ and *θ*_4_ are the angles between the propagation direction of the AE wave and the axial direction of the FBG AE sensors. The coordinate system is established on the surface of the thin PBX sample. The *Y* axis of the coordinate system is along the perpendicular direction passing through the center of FBG AE sensor 3. The *X* axis of the coordinate system is along the horizontal direction passing through the center of FBG AE sensor 4. Moreover, the origin O of the coordinate system is the intersection of the *X* axis and the *Y* axis. In the coordinate system, the center positions of the four FBG AE sensors are (*x*_1_,*y*_1_), (*x*_2_,*y*_2_), (*x*_3_,*y*_3_) and (*x*_4_,*y*_4_); the position of the broken lead is (*x_s_*,*y_s_*). Four sensors are placed radially clocked every 90 degrees in azimuth; this arrangement can guarantee that the broken lead signal in (*x_s_*,*y_s_*) is detected by all the sensors.

The point of the AE source in the traditional location method can be calculated as:(11)$$T_{1}-T_{2}=(\sqrt{{\left(x_{1}-x_{s}^{\prime }\right)}^{2}+{\left(y_{1}-y_{s}^{\prime }\right)}^{2}}-\sqrt{{\left(x_{2}-x_{s}^{\prime }\right)}^{2}+{\left(y_{2}-y_{s}^{\prime }\right)}^{2}})/v\;$$
(12)$$T_{1}-T_{3}=(\sqrt{{\left(x_{1}-x_{s}^{\prime }\right)}^{2}+{\left(y_{1}-y_{s}^{\prime }\right)}^{2}}-\sqrt{{\left(x_{3}-x_{s}^{\prime }\right)}^{2}+{\left(y_{3}-y_{s}^{\prime }\right)}^{2}})/v\;$$
(13)$$T_{1}-T_{4}=(\sqrt{{\left(x_{1}-x_{s}^{\prime }\right)}^{2}+{\left(y_{1}-y_{s}^{\prime }\right)}^{2}}-\sqrt{{\left(x_{4}-x_{s}^{\prime }\right)}^{2}+{\left(y_{4}-y_{s}^{\prime }\right)}^{2}})/v\;$$
where *T*_1_, *T*_2_, *T*_3_ and *T*_4_ are the arrival times for FBG AE sensor 1, sensor 2, sensor 3 and sensor 4, respectively. *v* is the velocity of the AE wave. $\left(x_{s}^{\prime },y_{s}^{\prime }\right)$ is the point of the AE source calculated by the traditional location method [27].

For the location experiment shown in Figure 10, *T*_1_ = 0.3060 ms, *T*_2_ = 0.3065 ms, *T*_3_ = 0.3800 ms, *T*_4_ = 0.3073 ms, *v* = 1145 (mm/ms), (*x*_1_*,y*_1_) = (100.0 mm,50.0 mm), (*x*_2_*,y*_2_) = (50.0 mm,100.0 mm), (*x*_3_*,y*_3_) = (0.0 mm,50.0 mm), and (*x*_4_*,y*_4_) = (50.0 mm,0.0 mm).

Calculating (11) and (12) by the least squares method in MATLAB software, the numerical error for the least square method is less than 1.0 × 10^−6^. The result is given by: (14)$$\left(x_{s1}^{\prime },y_{s1}^{\prime }\right)=\left(51.1\;\mathrm{mm},51.0\;\mathrm{mm}\right)\;\mathrm{units}\;\mathrm{not}\;\mathrm{italic}\;\mathrm{font}$$ where $\left(x_{s1}^{\prime },y_{s1}^{\prime }\right)$ is the point of the AE source calculated by (11) and (12).

Performing the same calculation for (12) and (13) as that described above, the result is:(15)$$\left(x_{s2}^{\prime },y_{s2}^{\prime }\right)=\left(51.1\;\mathrm{mm},50.3\;\mathrm{mm}\right)$$ where $\left(x_{s2}^{\prime },y_{s2}^{\prime }\right)$ is the point of the AE source calculated by (12) and (13).

Upon performing the same calculation of (11) and (13) as above, the result is: (16)$$\left(x_{s3}^{\prime },y_{s3}^{\prime }\right)=\left(50.5\;\mathrm{mm},51.1\;\mathrm{mm}\right)$$ where $\left(x_{s3}^{\prime },y_{s3}^{\prime }\right)$ is the point of the AE source calculated by (11) and (13).

Based on (14)–(16), the final average location of the AE source is:(17)$$\left(x_{s}^{\prime },y_{s}^{\prime }\right)=\left(50.9\;\mathrm{mm},50.8\;\mathrm{mm}\right)$$

The error *E*_1_ between $\left(x_{s},y_{s}\right)=\left(28.8\;\mathrm{mm},28.8\;\mathrm{mm}\right)$ and $\left(x_{s}^{\prime },y_{s}^{\prime }\right)=\left(50.9\;\mathrm{mm},50.8\;\mathrm{mm}\right)$ can be obtained, as shown in Figure 11. Figure 11 shows an enlarged view of the area defined by the dashed line in Figure 10:(18)$$\begin{array}{ll} E_{1} & =\sqrt{{\left(x_{s}-x_{s}^{\prime }\right)}^{2}+{\left(y_{s}-y_{s}^{\prime }\right)}^{2}}\\ & =\sqrt{{\left(28.8-50.9\right)}^{2}+{\left(28.8-50.8\right)}^{2}}\\ & =31.2\;\mathrm{mm} \end{array}$$

In Figure 11, there is a large error *E*_1_ between $\left(x_{s},y_{s}\right)$ and $\left(x_{s}^{\prime },y_{s}^{\prime }\right)$. The reason for this large error lies in the directionality of the FBG AE sensor. Therefore, the traditional location method (11)–(13) is not suitable for the FBG AE sensor; as a result, the location result obtained using the traditional method is inaccurate.

Based on the result of the FBG AE sensor directionality experiment, the time coefficient method is proposed to locate the source of the broken lead point. For the FBG AE sensor glued onto the surface of the thin PBX sample, the time coefficient *h*(*θ*) is introduced to calibrate the location. Based on (10), *h*(*θ*) can be described as:(19)$$\begin{array}{ll} h(\theta ) & =\frac{T(\theta =0{}^{\circ})}{T(\theta )}\\ & =\frac{2.3}{-2.623\times 10^{-8}\cdot \theta^{4}+1.429\times 10^{-5}\cdot \theta^{3}-4.542\times 10^{-4}\cdot \theta^{2}+1.314\times 10^{-2}\cdot \theta +2.3} \end{array}$$

In Figure 10, *θ*_1_, *θ*_2_, *θ*_3_ and *θ*_4_ can be calculated as follows:(20)$$\theta_{1}=\arctan\left(\frac{y_{1}-y_{s}}{x_{1}-x_{s}}\right)\;$$

(21)$$\theta_{2}=\arctan\left(\frac{x_{2}-x_{s}}{y_{2}-y_{s}}\right)\;$$

(22)$$\theta_{3}=\arctan\left(\frac{y_{3}-y_{s}}{x_{s}-x_{3}}\right)\;$$

(23)$$\theta_{4}=\arctan\left(\frac{x_{4}-x_{s}}{y_{s}-y_{4}}\right)\;$$

*h*(*θ*_1_), *h*(*θ*_2_), *h*(*θ*_3_) *and h*(*θ*_4_) can be obtained as follows:(24)$$h(\theta_{1})=\frac{2.3}{-2.623\times 10^{-8}\cdot \theta_{1}{}^{4}+1.429\times 10^{-5}\cdot \theta_{1}{}^{3}-4.542\times 10^{-4}\cdot \theta_{1}{}^{2}+1.314\times 10^{-2}\cdot \theta_{1}+2.3}\;$$(25)$$h(\theta_{2})=\frac{2.3}{-2.623\times 10^{-8}\cdot \theta_{2}{}^{4}+1.429\times 10^{-5}\cdot \theta_{2}{}^{3}-4.542\times 10^{-4}\cdot \theta_{2}{}^{2}+1.314\times 10^{-2}\cdot \theta_{2}+2.3}\;$$(26)$$h(\theta_{3})=\frac{2.3}{-2.623\times 10^{-8}\cdot \theta_{3}{}^{4}+1.429\times 10^{-5}\cdot \theta_{3}{}^{3}-4.542\times 10^{-4}\cdot \theta_{3}{}^{2}+1.314\times 10^{-2}\cdot \theta_{3}+2.3}\;$$(27)$$h(\theta_{4})=\frac{2.3}{-2.623\times 10^{-8}\cdot \theta_{4}{}^{4}+1.429\times 10^{-5}\cdot \theta_{4}{}^{3}-4.542\times 10^{-4}\cdot \theta_{4}{}^{2}+1.314\times 10^{-2}\cdot \theta_{4}+2.3}\;$$

Based on (19)–(27), the point of the AE source in the time coefficient location method can be given as follows:(28)$$T_{1}\cdot h(\theta_{1})-T_{2}\cdot h(\theta_{2})=(\sqrt{{\left(x_{1}-x_{s}^{\prime \prime }\right)}^{2}+{\left(y_{1}-y_{s}^{\prime \prime }\right)}^{2}}-\sqrt{{\left(x_{2}-x_{s}^{\prime \prime }\right)}^{2}+{\left(y_{2}-y_{s}^{\prime \prime }\right)}^{2}})/v\;$$
(29)$$T_{1}\cdot h(\theta_{1})-T_{3}\cdot h(\theta_{3})=(\sqrt{{\left(x_{1}-x_{s}^{\prime \prime }\right)}^{2}+{\left(y_{1}-y_{s}^{\prime \prime }\right)}^{2}}-\sqrt{{\left(x_{3}-x_{s}^{\prime \prime }\right)}^{2}+{\left(y_{3}-y_{s}^{\prime \prime }\right)}^{2}})/v\;$$
(30)$$T_{1}\cdot h(\theta_{1})-T_{4}\cdot h(\theta_{4})=(\sqrt{{\left(x_{1}-x_{s}^{\prime \prime }\right)}^{2}+{\left(y_{1}-y_{s}^{\prime \prime }\right)}^{2}}-\sqrt{{\left(x_{4}-x_{s}^{\prime \prime }\right)}^{2}+{\left(y_{4}-y_{s}^{\prime \prime }\right)}^{2}})/v\;$$
where *h*(*θ*_1_), *h*(*θ*_2_), *h*(*θ*_3_) *and h*(*θ*_4_) are the time coefficients of *T*_1_, *T*_2_, *T*_3_ and *T*_4_, respectively. $\left(x_{s}^{\prime \prime },y_{s}^{\prime \prime }\right)$ is the point of the AE source calculated by the time coefficient location method.

Calculating (28) and (29) by the least squares method in MATLAB software, the numerical error for the least squares method is less than 1.0 × 10^−6^; the result is given as follows:(31)$$\left(x_{s1}^{\prime \prime },y_{s1}^{\prime \prime }\right)=\left(29.3mm,30.1\;\mathrm{mm}\right)$$ where $\left(x_{s1}^{\prime \prime },y_{s1}^{\prime \prime }\right)$ is the point of the AE source calculated by (28) and (29).

Performing the same calculation of (29) and (30) as that described above, the result is
(32)$$\left(x_{s2}^{\prime \prime },y_{s2}^{\prime \prime }\right)=\left(30.1\;\mathrm{mm},29.8\;\mathrm{mm}\right)$$
where $\left(x_{s2}^{\prime \prime },y_{s2}^{\prime \prime }\right)$ is the point of the AE source calculated by (29) and (30).

Performing the same calculation of (28) and (30) as that described above, the result is
(33)$$\left(x_{s3}^{\prime \prime },y_{s3}^{\prime \prime }\right)=\left(29.9\;\mathrm{mm},30.8\;\mathrm{mm}\right)$$
where $\left(x_{s3}^{\prime \prime },y_{s3}^{\prime \prime }\right)$ is the point of the AE source calculated by (28) and (30).

Based on (31)–(33), the average location of the AE source is
(34)$$\left(x_{s}^{\prime \prime },y_{s}^{\prime \prime }\right)=\left(29.8\;\mathrm{mm},30.2\;\mathrm{mm}\right)\;$$

The error *E*_2_ between $\left(x_{s},y_{s}\right)=\left(28.8\;\mathrm{mm},28.8\;\mathrm{mm}\right)$ and $\left(x_{s}^{\prime \prime },y_{s}^{\prime \prime }\right)=\left(29.8\;\mathrm{mm},30.2\;\mathrm{mm}\right)$ can be obtained, as shown in Figure 11.(35)$$\begin{array}{ll} E_{2} & =\sqrt{{\left(x_{s}-x_{s}^{\prime \prime }\right)}^{2}+{\left(y_{s}-y_{s}^{\prime \prime }\right)}^{2}}\\ & =\sqrt{{\left(28.8-29.8\right)}^{2}+{\left(28.8-30.2\right)}^{2}}\\ & =1.7\;\mathrm{mm} \end{array}$$

In Figure 11, error *E*_2_ is less than error *E*_1_. For the FBG AE sensor glued onto the surface of PBX, the time coefficient location method of (28)–(30) is clearly better than the traditional location method of (11)–(13).

The experiments were repeated six times at different positions, as shown in Figure 12.

In Figure 12, No. 1, No. 2, No. 3, No. 4, No. 5 and No. 6 are the locations of the broken lead experiments. In the coordinate system, the locations of No. 1, No. 2, No. 3, No. 4, No. 5 and No. 6 are (28.8, 28.8), (35.9, 35.9), (42.9, 42.9), (40.0, 50.0), (30.0, 50.0) and (20.0, 50.0), respectively.

The arrival time of the four FBG AE sensors recorded in the experiments repeated six times are shown in Table 2.

The results of the location calculations are shown in Table 3.

In Table 3, the error *E*_1_ is dependent on the position, i.e., lower error for position No. 3 and higher error for position No. 6. The different results for the different positions correspond to different *θ*_1_, *θ*_2_, *θ*_3_ and *θ*_4_, as shown in Figure 10b. The greater the deviation of the four angles is, the greater is the influence of the FBG AE sensor directivity, and the greater the final error.

In Table 3, the location errors *E*_1_ calculated by the traditional location method are 31.2 mm, 4.5 mm, 2.5 mm, 9.5 mm, 19.0 mm, and 55.0 mm. The location errors *E*_2_ calculated by the time coefficient method are 1.7 mm, 1.2 mm, 1.7 mm, 1.4 mm, 1.7 mm, and 1.3 mm. Obviously, the location errors of the time coefficient method are much smaller than those of the traditional location method.

## 4. Summary and Conclusions

In this paper, the novelty lies in three areas. The first is the time coefficient location method based on the directionality of the FBG AE sensor. The second is the establishment of a four-channel FBG AE device to perform the experiments. The last is FBG AE sensor experiments conducted on thin PBX materials.

In the FBG AE sensor directional experiment, the amplitude and arrival time detected by the FBG AE sensor were found to be related to the angle between the propagation direction of the AE wave and the axial direction of the FBG AE sensors. Based on the directionality of the FBG AE sensor, the time coefficient location method was proposed to correct the time detected by the FBG AE sensors. The results of the location experiments on the surface of thin PBX showed that the accuracy of the sensor is good.

For the FBG AE sensors mentioned in the paper, the threshold voltage is 40 dB. As a result, smaller actual defects/cracks could not be detected. Only when the amplitude of an actual crack is greater than 40 dB can the AE wave be collected by the sensors. For the time coefficient location method, only the surface cracks limited to the region near the center of thin PBX could be located. Otherwise, the method must be changed. Thus, we believe that the FBG AE sensors and the time coefficient location method need further research before their fully mature application in tasks involving PBX materials. In the paper, usage of a multiple FBG-based system for AE measurements on PBX increases the complexity and installation and equipment costs. However, it could also improve the safety of PBX materials, which are the main component of detonation devices in nuclear bombs and the main part of explosive bolts for multistage heavy launch vehicles and large satellite solar wings. Therefore, any investment in the safety of PBX materials would help improve the safety of those important strategic weapons.

## Figures and Tables

**Figure 1 sensors-18-03778-f001:**
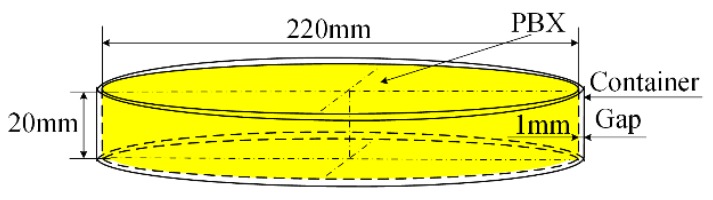
Thin PBX in a limited space container.

**Figure 2 sensors-18-03778-f002:**
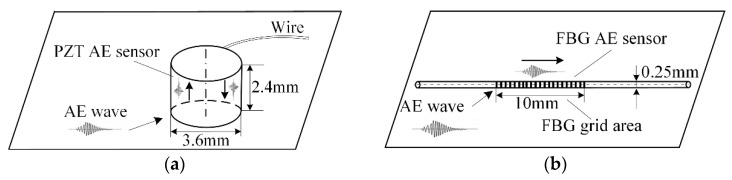
PZT and FBG AE sensor. (**a**) PZT AE sensor; (**b**) FBG AE sensor.

**Figure 3 sensors-18-03778-f003:**
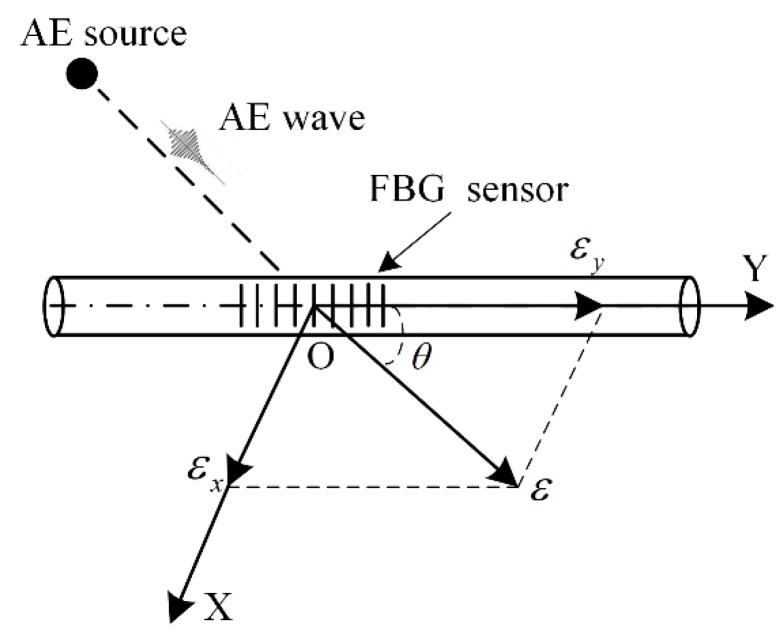
The coordinate system of the FBG sensor and the strain decomposition.

**Figure 4 sensors-18-03778-f004:**
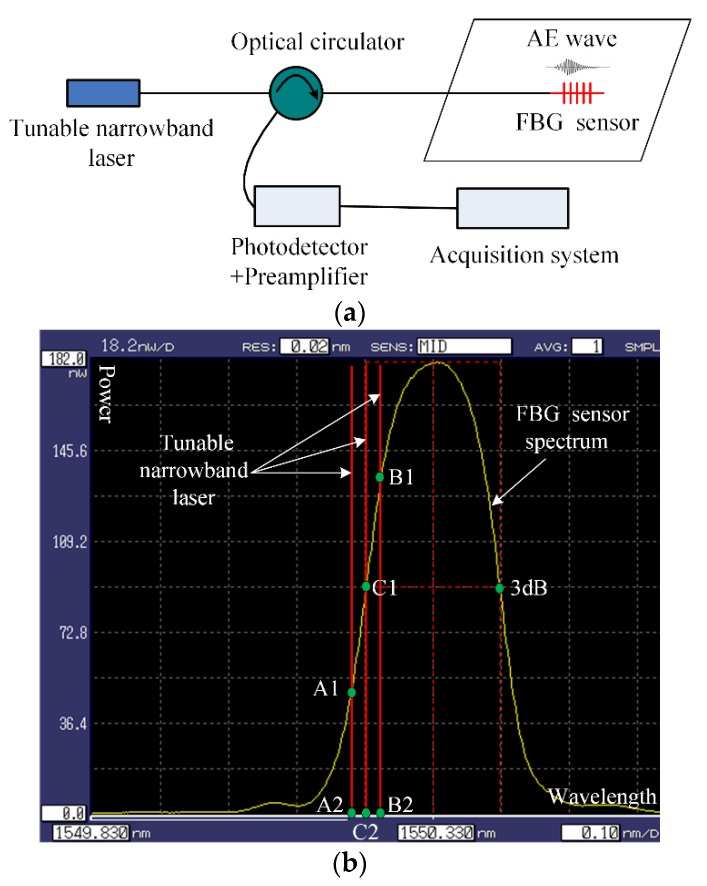
Tunable narrowband laser detection system and its illustrative diagram. (**a**) Single channel FBG sensor system; (**b**) Illustrative diagram.

**Figure 5 sensors-18-03778-f005:**
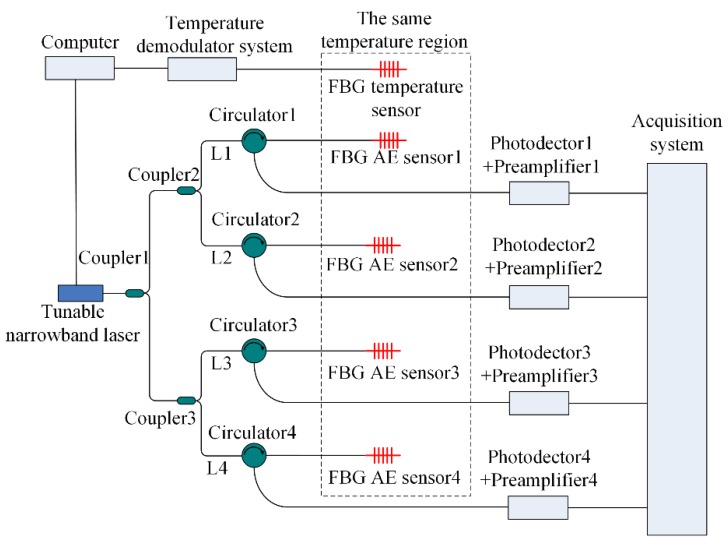
Four-channel FBG AE sensor system.

**Figure 6 sensors-18-03778-f006:**
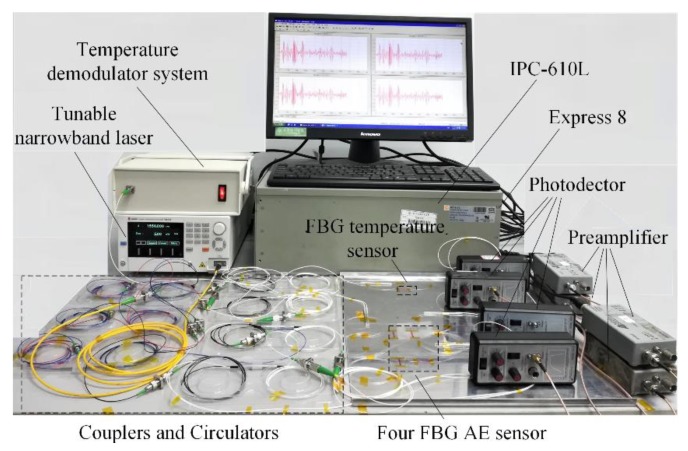
FBG AE system setup.

**Figure 7 sensors-18-03778-f007:**
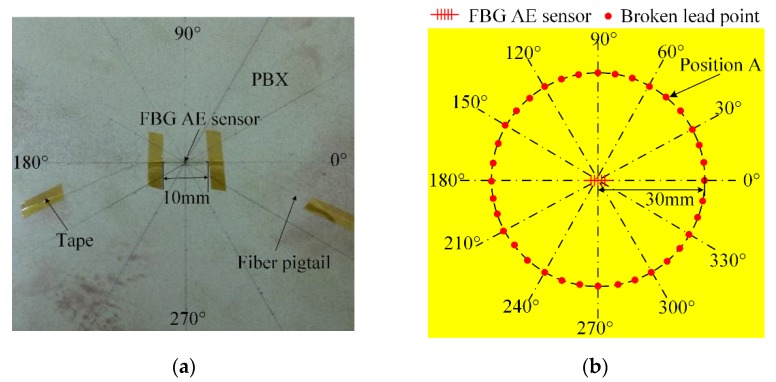
FBG AE sensor directionality experiment. (**a**) Photograph of the FBG AE sensor on the PBX surface; (**b**) FBG AE sensor experiment and broken lead point.

**Figure 8 sensors-18-03778-f008:**
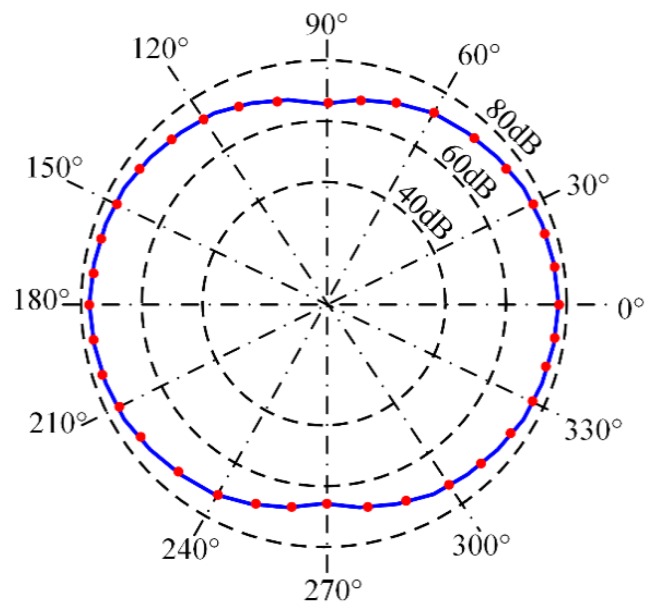
Polar diagram of the FBG AE sensor signal amplitude.

**Figure 9 sensors-18-03778-f009:**
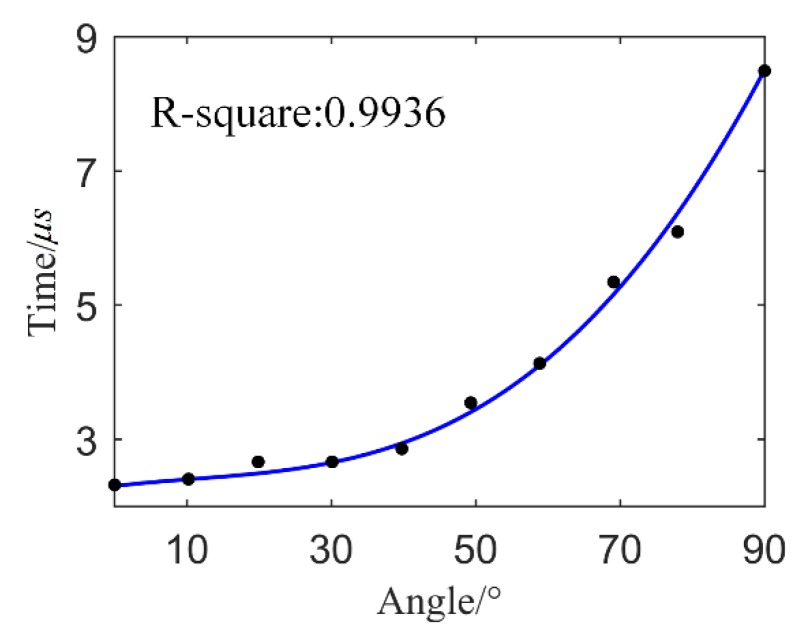
Polynomial function *T*(*θ*).

**Figure 10 sensors-18-03778-f010:**
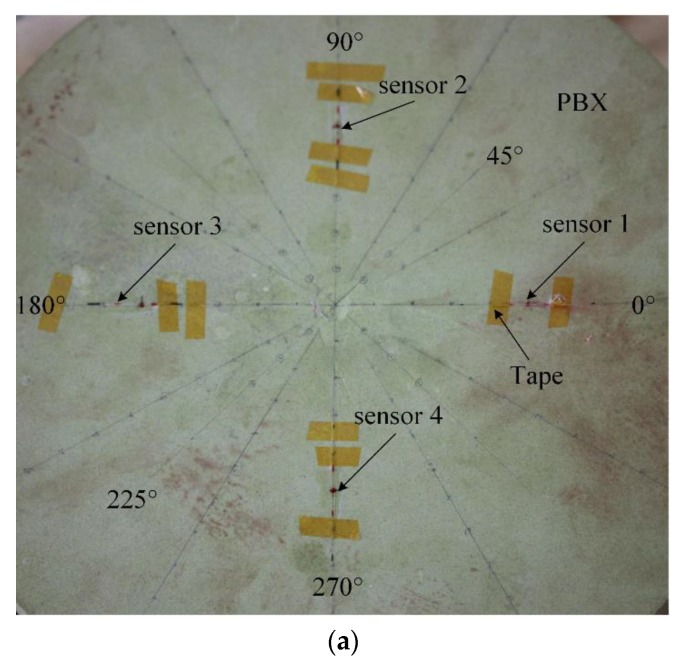
FBG AE source location experiment on a thin PBX sample. (**a**) Photograph of four sensors on the PBX surface; (**b**) source location experiment of a broken lead point.

**Figure 11 sensors-18-03778-f011:**
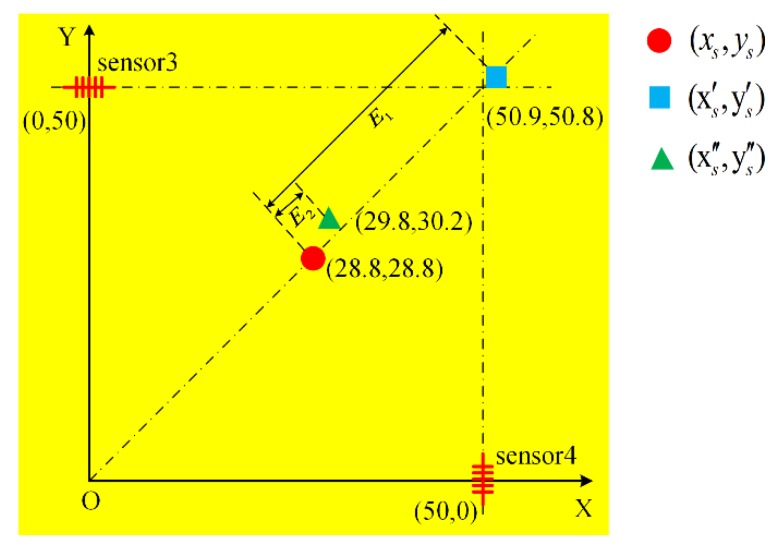
The result of the location experiment.

**Figure 12 sensors-18-03778-f012:**
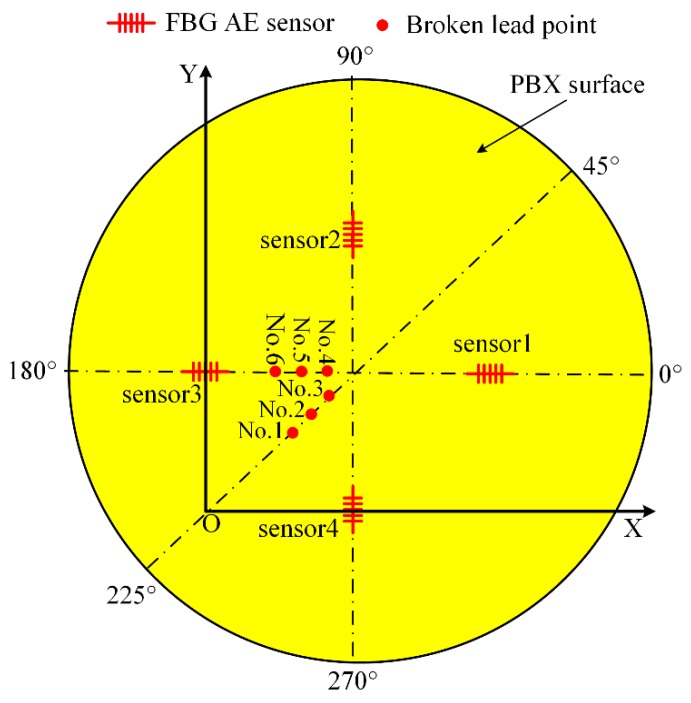
Locations of the broken lead.

**Table 1 sensors-18-03778-t001:** Angle *θ* and arrival time *T* of the FBG AE sensor.

Angle *θ*	Arrival Time *T*/μs
0°	2.30
10°	2.40
20°	2.60
30°	2.65
40°	2.90
50°	3.55
60°	4.20
70°	5.40
80°	6.20
90°	8.50

**Table 2 sensors-18-03778-t002:** Arrival time.

Experiment Locations	T_1_/ms	T_2_/ms	T_3_/ms	T_4_/ms
No. 1	0.3360	0.3365	0.3380	0.3373
No. 2	0.2065	0.2080	0.1890	0.1875
No. 3	0.4750	0.4755	0.4660	0.4653
No. 4	0.3602	0.3700	0.3505	0.3703
No. 5	0.2505	0.2680	0.2365	0.2665
No. 6	0.3468	0.3790	0.3003	0.3770

**Table 3 sensors-18-03778-t003:** Location results and their errors.

Experiment Locations	$\;\left(x_{s},y_{s}\right)$ /mm	$\;\left(x_{s}^{\prime },y_{s}^{\prime }\right)$ /mm	*E*_1_ /mm	$\;\left(x_{s}^{\prime \prime },y_{s}^{\prime \prime }\right)$ /mm	*E*_2_ /mm
No. 1	(28.8,28.8)	(50.9,50.8)	31.2	(29.8,30.2)	1.7
No. 2	(35.9,35.9)	(39.8,38.0)	4.5	(35.0,36.6)	1.2
No. 3	(42.9,42.9)	(44.9,44.4)	2.5	(44.3,43.9)	1.7
No. 4	(40.0,50.0)	(49.5,50.0)	9.5	(39.5,48.7)	1.4
No. 5	(30.0,50.0)	(48.8,47.0)	19.0	(29.7,48.4)	1.7
No. 6	(20.0,50.0)	(70.4,71.7)	55.0	(21.2,49.5)	1.3

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
