# Peer review of "Application of Fiber Bragg Grating Acoustic Emission Sensors in Thin Polymer-Bonded Explosives"

_sensors, 2018, doi:10.3390/s18113778_

Round 1

Reviewer 1 Report

The work has improved with respect the previous version. It is fine for me

Author Response

Dear reviewer,

Thanks for your time and patience.

Reviewer 2 Report

The paper 379795 entitled "Application of Fiber Bragg Grating Acoustic Emission Sensors in Thin Polymer-Bonded Explosives" is a resubmission from paper 349069.

Compared to the previous versions, the improvement is significant but the paper still need some corrections:

Minor corrections:

1. L76 and L90: refraction index would be better called refractive index

2. L184:  "... the angle from 0° to 350° ..." ==> "... to 330°"

3. Ref.16 and ref.124 are identical

Major correction: see attached file

Author Response

Dear reviewer,

Thanks for your time and patience. The answers are in the attachment file.

Round 2

Reviewer 2 Report

Paper is now suitable for publication